# Evolutionary Insight into the Association between New Jersey Polyomavirus and Humans

**DOI:** 10.3390/v15112248

**Published:** 2023-11-13

**Authors:** Aref-Abdolllah Aghebatrafat, Chris Lauber, Kevin Merkel, Barbara Fruth, Kevin Langergraber, Martha M. Robbins, Roman M. Wittig, Fabian H. Leendertz, Sébastien Calvignac-Spencer

**Affiliations:** 1Epidemiology of Highly Pathogenic Microorganisms, Robert Koch Institute, 13353 Berlin, Germany; arefrafat@gmail.com (A.-A.A.); merkelk@rki.de (K.M.); fabian.leendertz@helmholtz-hioh.de (F.H.L.); 2Institute for Experimental Virology, TWINCORE Centre for Experimental and Clinical Infection Research, a Joint Venture between Medical School Hannover (MHH) and Helmholtz Centre for Infection Research (HZI), 30625 Hannover, Germany; chris.lauber@twincore.de; 3Cluster of Excellence 2155 RESIST, 30625 Hannover, Germany; 4Max-Planck-Institute of Animal Behavior, 78467 Konstanz, Germany; bfruth@ab.mpg.de; 5Centre for Research and Conservation/KMDA, B-2018 Antwerp, Belgium; 6Faculty of Science, School of Biological and Environmental Sciences, Liverpool John Moores University, Liverpool L3 3AF, UK; 7School of Human Evolution and Social Change and Institute of Human Origins, Arizona State University, Tempe, AZ 85281, USA; kevin.langergraber@asu.edu; 8Departement of Primate Behavior and Evolution, Max Planck Institute for Evolutionary Anthropology, 04103 Leipzig, Germany; robbins@eva.mpg.de; 9Taï Chimpanzee Project, Centre Suisse de Recherches Scientifiques, Abidjan 1303, Côte d’Ivoire; roman.wittig@isc.cnrs.fr; 10The Ape Social Mind Lab, Institut des Sciences Cognitives, CNRS UMR 5229, 69500 Bron, France; 11Helmholtz Institute for One Health, Helmholtz-Centre for Infection Research (HZI), 17489 Greifswald, Germany; 12Faculty of Mathematics and Natural Sciences, University of Greifswald, 17489 Greifswald, Germany

**Keywords:** New Jersey polyomavirus, virus evolution, codivergence, hominine, African great apes

## Abstract

Advances in viral discovery techniques have led to the identification of numerous novel viruses in human samples. However, the low prevalence of certain viruses in humans raises doubts about their association with our species. To ascertain the authenticity of a virus as a genuine human-infecting agent, it can be useful to investigate the diversification of its lineage within hominines, the group encompassing humans and African great apes. Building upon this rationale, we examined the case of the New Jersey polyomavirus (NJPyV; *Alphapolyomavirus terdecihominis*), which has only been detected in a single patient thus far. In this study, we obtained and analyzed sequences from closely related viruses infecting all African great ape species. We show that NJPyV nests within the diversity of these viruses and that its lineage placement is compatible with an ancient origin in humans, despite its apparent rarity in human populations.

## 1. Introduction

NJPyV was initially discovered in a transplant recipient who had developed retinal blindness and vasculitic myopathy [1]. Since then, its genetic material has not been detected in any human samples. For example, a recent search into >143,000 metagenomic Sequence Read Archives at NCBI identified >450 experiments containing PyV reads, none matching NJPyV sequences [2]. Serological studies have also found that few humans have antibodies that react to NJPyV [3,4]. As a result, some authors have suggested that NJPyV may not be a human-adapted virus [2,4].

However, the closest known relatives of NJPyV are PyVs that infect nonhuman primates, as is also the case for five human-adapted PyVs. The closest genomes in public databases were identified in chimpanzees (*Pan troglodytes*), and multiple PyV lineages infecting captive chimpanzees belonging to three subspecies: Western chimpanzee [*P. t. verus*], Central chimpanzee [*P. t. troglodytes*], and Eastern chimpanzee [*P. t. schweinfurthii*]) [5,6]. These PyV lineages exhibit some degree of host specificity, supporting the idea of a long-term association with their chimpanzee hosts. In this study, we aim to formally test the hypothesis of a long shared evolutionary history of NJPyV-related viruses and hominines (African great apes and humans) and, consequently, assess the possibility of NJPyV being an ancient associate of mankind.

## 2. Materials and Methods

### 2.1. Samples

Fecal and urine samples were collected from habituated and unhabituated great apes from all 4 species of African great apes and 7 of the 9 subspecies (Table 1). We expected intermittent detection from infected individuals; thus, fecal samples collected from habituated great apes were selected randomly. Accordingly, the number of infected individuals is lower than the number of positive samples, whose proportion cannot be used to infer prevalence in the population. Fecal samples collected from nonhabituated animals were not genotyped so the same limitation applies. Fecal samples were preserved in a nucleic acid preservation buffer [7] and either flash frozen or kept at ambient temperature depending on field sites. Urine samples were flash frozen.

### 2.2. DNA Extraction, PCR Screening and Sequencing

DNA was extracted from a pea-size subsample using the GeneMATRIX Stool DNA Purification kit (EURx Ltd., Gdansk, Poland) according to manufacturer’s instructions.

We first ran a confirmatory PCR targeting a short fragment of the mitochondrial 16S gene to confirm the host species [8].

We then implemented an in-house semi-nested PCR designed to amplify a ca. 170 bp-long VP1 fragment in NJPyV-related viral genomes. Out of the three primers (NJPyV-f1 5′-GAAATGARAATTCTAGRTATTWTGGMTC-3′, NJPyV-r1 5′-GCYYCATTRTCTGCCTTTACCA-3′ and NJPyV-r2 5′-CCTTTACCAWRACHCCACADATATC-3′), the two reverses were very specific to NJPyVr sequences in preliminary BLAST analyses restricted to members of the *Polyomaviridae* family. This system was validated on DNA extracts from spleen and liver necropsies of a captive gorilla kept at the Limbe Wildlife Center (Cameroon) that had previously been found to be NJPyV-related virus-positive (personal communication Bernhard Ehlers).

PCR reactions of 25 µL were set up to contain 0.2 µM of each primer, 200 µM dNTP (with dUTP replacing dTTP), 1.25 U Platinum^®^Taq polymerase (InvitrogenTM, Waltham, MA, USA), 4 mM MgCl_2_ and 1× PCR buffer; 3 µL fecal DNA was tested in first-round reactions; 1 µL of first-round reaction diluted 40 times was used to seed second-round reactions. Cycling conditions were as follows: 5 min at 95 °C, followed by 45 cycles of 30 s at 95 °C, 30 s at 56 °C (first round) or 58 °C (second round), and 45 s at 72 °C, with a final 10 min at 72 °C.

PCR products were purified using ExoSAP-IT (Affymetrix, Santa Clara, CA, USA) and sequenced with a BigDye Terminator cycle sequencing kit on a DNA automated sequencer (Applied Biosystems, Warrington, UK). The resulting short VP1 sequences are available as Appendix A.

We tried to obtain longer NJPyV-related viral sequences using in-solution hybridization capture with baits covering the genomes of 75 PyV species recognized by the ICTV (as of 2016), including the reference genome for NJPyV (NC_024118) (Daicel Arbor Biosciences, Ann Arbor, MI, USA). All attempts failed, including when using the relatively good-quality DNA extracts derived from the positive control. Since full genomes from contaminant PyVs were often recovered (e.g., Merkel cell polyomavirus), we hypothesize that this failure was due to exceedingly low quantities of NJPyV-related viral genetic material in the starting material.

### 2.3. Short Read Archive Mining

We analyzed 26,788 raw, unprocessed sequencing datasets from great apes (Hominidae) available in the Sequence Read Archive (SRA) by July 2020. These sequencing experiments were subjected to a profile Hidden Markov Model (pHMM)-based sequence homology search using a multiple sequence alignment of Large T antigen proteins of reference polyomaviruses as query. For the hits obtained, we performed a seed-based, targeted viral genome assembly using Genseed-HMM [9]. The SRA-based computational virus discovery workflow is described in previous studies [10,11] and an implementation is provided by the Virushunter and Virusgatherer tools, which are available at https://github.com/lauberlab/VirusHunterGatherer (accessed on 1 September 2023). Virushunter and Virusgatherer were run in a highly parallelized fashion on the high-performance computing cluster Taurus of the Dresden University of Technology (TU). The assembled polyomavirus sequences were manually validated and circularized in case they represented full-length viral genomes. The NJPyV-related viral genome is available as Appendix A.

### 2.4. Phylogenetic Analyses

To assemble a reference dataset, we first downloaded the 13,183 sequences assigned to the family *Polyomaviridae* in NCBI (as of September 2022). From those, we identified 203 (including 26 genomes) for which an African great ape (AGA) was identified as the host. One hundred forty-eight accessions were or comprised VP1 sequences, on which we focused since we had only generated short sequences in this region of the genome.

We then ran Fabox to identify unique sequences [12]. Ninety-two were found. Only three were associated with more than one host species/subspecies (in all cases, distinct chimpanzee subspecies). They had been obtained in a study on captive animals, possibly kept with representatives of other subspecies [6]. We added sequences extracted from the reference genomes of the 117 PyV species currently recognized by the International Committee for the Taxonomy of Viruses (https://ictv.global/report/chapter/polyomaviridae/polyomaviridae, accessed on 1 September 2023) to these AGA PyV VP1 sequences, as well as that of the recently described Quebec PyV [2].

These 210 reference sequences were combined with the 13 short sequences obtained from our PCR screening and the 1 sequence recovered from existing high-throughput sequencing (HTS) data. They were then aligned at the amino acid level using muscle [13], as implemented in SeaView v5 [14]. After removing poorly conserved positions in this alignment using gBlocks [15], we ran an exploratory phylogenetic analysis using IQTree2 [16] with automated model selection and estimating branch robustness with Shimodaira–Hasegawa-like approximate likelihood ratio tests [17]. This analysis confirmed that all sequences generated in our study and the single sequence retrieved from HTS data were among the closest relatives to NJPyV, together with 24 AGA-derived PyV sequences already available in public databases (SH-like aLRT: 97.8). These 37 NJPyV/NJPyV-related viral sequences were not all unique and did not all overlap. The final sequence dataset that we assembled comprised 13 unique sequences that overlapped with the VP1 region targeted by our screening PCR (Appendix A).

We first ran an exploratory Bayesian Markov chain Monte Carlo analysis with BEAST v1.10.4 [18], using an uncalibrated uncorrelated relaxed clock with a lognormal distribution. For this and all following models we used: i. the nucleotide substitution model selected by jModelTest 2.1.10 [19] using the Bayesian information criterion, i.e., HKY with rate variation across sites; ii. a birth–death speciation prior, following Ritchie, Lo and Ho’s suggestion that it produces more reliable date estimates than a Yule prior from datasets representing inter- and intraspecific sampling [20]. We ran two 10 million generation chains for each model and assessed their convergence and mixing behavior using Tracer v1.7 [21]. We also used the latter software to assess burn-in and effective sample sizes (ESS) for all parameters of our models, aiming at ESS > 200 for a model’s combined runs. We summarized posterior samples of trees as maximum clade credibility (MCC) trees using RootAnnotator, a tool that essentially implements TreeAnnotator (distributed with BEAST) while also computing root posterior probabilities [22].

We then built constrained models to assess specific evolutionary hypotheses.

We first ran two uncalibrated models for which root position was constrained as follows: (i) ((NJPyV, NJPyV-related viruses from gorilla+Eastern chimpanzee), (NJPyV-related viruses from all other panines)) (non-codivergent model); or (ii) ((NJPyV, NJPyV-related viruses from all other panines), NJPyV-related viruses from gorilla+Eastern chimpanzee) (codivergent model). The aim was to assess whether the data decisively supported one of these roots over the other without any other constraint. We found that the non-codivergent model had a higher marginal likelihood but was not decisively better than the codivergent model (2lnBF = 4.8; for details about model comparison, please see the end of this section).

We also ran two uncalibrated models for which additional constraints were implemented on the non-codivergent and codivergent models, enforcing monophyly on NJPyV-related viruses in gorilla + Eastern chimpanzees and NJPyV-related viruses in other chimpanzees. These models provided us with the background against which to compare calibrated models, reducing phylogenetic uncertainty where branch robustness suggested it was minimal.

Finally, we ran a series of calibrated models for which node ages were modeled to match specific evolutionary scenarios and, therefore, specific host divergence dates with which they were assumed to be synchronous. We used host divergence date estimates derived from two studies [23,24], running analyses with one or the other set of calibrations. These estimates and how we modeled them are summarized in Table 2. An exemplary XML file detailing the set-up of the model assuming codivergence and using the Perelman calibration scheme is available as Appendix A.

The non-codivergent models that assumed the root ((NJPyV, NJPyV-related viruses from gorilla + Eastern chimpanzee), (NJPyV-related viruses from all other panines)) all used *Pan* and *Pan troglodytes* calibrations but differed with respect to the modeling of a single older node age. The models that assumed a transmission from the gorilla to the human lineage were further calibrated with a root age corresponding to the *Homo + Pan + Gorilla* divergence date; the models that assumed transmission from the human to the gorilla lineage with a root age corresponding to the *Homo + Pan* divergence date; and the models that assumed incomplete lineage sorting with a (NJPyV, NJPyV-related viruses from gorilla+Eastern chimpanzee) time to the most recent common ancestor (tMRCA) corresponding to the *Homo + Pan + Gorilla* divergence date. The codivergent models that assumed the ((NJPyV, NJPyV-related viruses from all other panines), (NJPyV-related viruses from gorilla+Eastern chimpanzee)) root were calibrated with the *Pan* and *Pan troglodytes* calibrations as well as with a root age corresponding to the *Homo + Pan + Gorilla* divergence date and a (NJPyV, NJPyV-related viruses from all other panines) tMRCA corresponding to the *Homo + Pan* divergence date.

To compare model performance, we estimated marginal likelihoods using stepping stone sampling (50 steps of 500 k generations) [25]. We interpreted differences in marginal likelihood estimates (MLE) as described by Kass and Raftery (1995), considering that 2ln(B10) below 2 was barely worth mentioning, between 2 and 6 showed positive evidence against H0, between 6 and 10 showed strong evidence against H0, and above 10 showed decisive evidence against H0 [26].

The tanglegram in Figure 1 was generated with the function *cophylo* of the package *phytools* in R v4.3.1 [27,28]. Dates for the host phylogeny were derived from Timetree [29], except for dates within *Pan troglodytes*, which were derived from [30]. We did not use Timetree-derived dates for formal calibration as these are aggregate dates that may not all originate in the same original studies. The phylogenetic trees in Figure 2 were prepared with iTol v5 [31].

## 3. Results

### 3.1. PCR Screening and Data Mining

Using a NJPyV-related virus-specific PCR system, we screened 276 fecal and 53 urine samples collected from an unknown number of wild individuals of all four species of African great apes (AGA; seven of the nine subspecies) (Table 1). Short VP1 sequences derived from the resulting amplicons were used for preliminary phylogenetic analyses, which confirmed the detection of NJPyV-related viruses in four fecal samples from Western chimpanzees (three individuals), two from Eastern chimpanzees (at least one individual), and seven from bonobos (*P. paniscus*; at least three individuals). We also screened short read archives from AGA sequencing experiments and identified NJPyV-related viral sequences in a lowland gorilla (*Gorilla gorilla*) sample (original sequencing file: SRR8674971). In addition, we collected all AGA PyV sequences from the nonredundant database of the National Center for Biotechnology Information (NCBI, n = 148), identifying eight NJPyV-related viral sequences associated with Western chimpanzees, eight with Central chimpanzees, three with Eastern chimpanzees, one with bonobos, two with lowland gorillas, and two with mountain gorillas (*G. beringei*).

### 3.2. Phylogenetic Analyses

To investigate the evolutionary history of these viruses, we ran Bayesian analyses using BEAST v1.10 [18]. We focused on an ingroup dataset of 13 unique partial and complete VP1 gene sequences, including NJPyV and NJPyV-related viruses from Western, Central, and Eastern chimpanzees, bonobos, and lowland gorillas. In a first (uncalibrated) phylogenetic analysis, NJPyV-related viral sequences from Western and Central chimpanzees as well as from bonobos formed moderately to well-supported clades. Putative codivergence events were clearly identifiable in this part of the tree, which recapitulated the host phylogeny (Figure 1). In contrast, Eastern chimpanzee NJPyV-related viral sequences nested within the diversity of lowland gorilla NJPyV-related viral sequences (posterior probability: 1.00), possibly indicating a transmission event from gorillas to chimpanzees, as already observed in other families of viruses with a dsDNA genome [32,33].

Intriguingly, NJPyV formed a deep-branching lineage sister to the putative gorilla NJPyV-related viral lineage (0.81) (Figure 1). This topology is in apparent contradiction with codivergence at the host genus level and might reflect: (i) cross-species transmission from gorillas to humans; (ii) from humans to gorillas; or (iii) incomplete viral lineage sorting [34] (15.5% of >500 bp sequences in hominine genomes show this topology [35]). Yet we also noted that the second best-supported root was compatible with codivergence at the host genus level, with NJPyV forming a clade with NJPyV-related viruses from chimpanzees and bonobos (0.11). Bayes factor comparison favored the best-supported topology but did not allow for a decisive rejection of host genus-level codivergence (2lnBF = 3.8; <10; Table 3).

### 3.3. Molecular Clock Analyses

All the aforementioned scenarios imply slightly different timescales. We compared calibrated molecular clock models reflecting these hypotheses, imposing topological constraints and divergence dates in the virus tree that matched divergence events in the host tree, using estimates from two independent studies (Table 2; [23,24]). Models assuming codivergence, cross-species transmission from gorillas to humans, and incomplete viral lineage sorting all had similar marginal likelihoods (2 lnBF < 2) and were all inferior to the model assuming transmission from humans to gorillas, although not decisively so (2.8 < 2ln BF < 4.2; Table 3).

While all calibrated models resulted in a drop in marginal likelihood with respect to their uncalibrated counterparts, the codivergence model and the model assuming transmission from humans to gorillas were the only models not to be decisively rejected (minimum 2lnBF = −7.2 and −9.0, respectively; >−10; Figure 2A,B). In addition, and irrespective of the calibration scheme, the date of divergence of the NJPyV and gorilla NJPyV-related viral lineages was estimated to lie >4 million years ago (Table 3), suggesting that even a gorilla to human transmission scenario would imply millions of years of evolution in human hosts.

## 4. Discussion

We searched for NJPyV-related viruses in wild African great apes, which we detected in multiple populations of Western and Eastern chimpanzees and bonobos. Combined with the identification of another NJPyV-related virus in high-throughput sequencing data from a lowland gorilla and earlier reports of NJPyV-related viral sequences from captive individuals of multiple African great ape species/subspecies [8,9], this suggests that NJPyV-related viruses naturally infect most, if not all, members of this primate lineage.

The long-term evolution of polyomaviruses is characterized by stable associations with their hosts, which translate into frequent codivergence events [36,37,38,39,40]. In this context, NJPyV/NJPyV-related viruses form a well-supported monophyletic group whose closest relatives infect other nonhuman primates, which is consistent with a local pattern of codivergence with their primate hosts [39]. In line with this observation, we identified multiple examples of putative codivergence events within the NJPyV/NJPyV-related virus clade, extending and reinforcing earlier findings [9]. Accordingly, it seems reasonable to assume that the natural host of NJPyV is a hominine.

Since the best-supported placement of NJPyV did not immediately suggest a simple pattern of codivergence (and therefore did not immediately suggest that humans are NJPyV natural hosts), we explored alternative evolutionary scenarios using (molecular clock) model comparisons in a Bayesian framework. Importantly, the two molecular clock models not firmly rejected in these analyses both assumed that the NJPyV lineage has infected the human lineage ever since its divergence from African great ape lineages. We conclude that the patterns of genetic diversity of NJPyV/NJPyV-related viruses are, at this stage, compatible with NJPyV being a human-infecting agent. Importantly, this aligns well with the NJPyV patient history since this person did not report recent international travel nor have contact with domestic animals or nonhuman primates [1]. We acknowledge that further sampling of hominine- and other mammal-infecting PyV may alter the patterns we observed. However, we do not see this eventuality as very likely, considering the already consequent search effort in African great apes.

If one accepts the notion that NJPyV is a human-infecting virus, the apparent absence of NJPyV from human populations remains a conundrum. It seems unlikely that NJPyV would currently be on the brink of extinction in humans, given the ability of its close relatives in African great apes to survive in much smaller fragmented populations. Plausible explanations might include: (i) that NJPyV replication levels in healthy individuals are extremely low, consistent with undetectable viral load in a pretransplant serum sample of the NJPyV patient [1]; (ii) that its main site of replication has been poorly sampled and tested (NJPyV exhibited a broad vascular endothelial cell tropism in the only known case, but whether this reflects its tropism in healthy individuals is unknown [1]); and/or (iii) that it has an uneven global distribution and is more prevalent in (or even nearly restricted to) poorly sampled regions of the world, like several recently discovered lineages of pathogens [41,42]. We hypothesize that NJPyV and possibly other human-adapted viruses rarely associated with disease will be detected in the future, as such biases are progressively corrected.

## Figures and Tables

**Figure 1 viruses-15-02248-f001:**
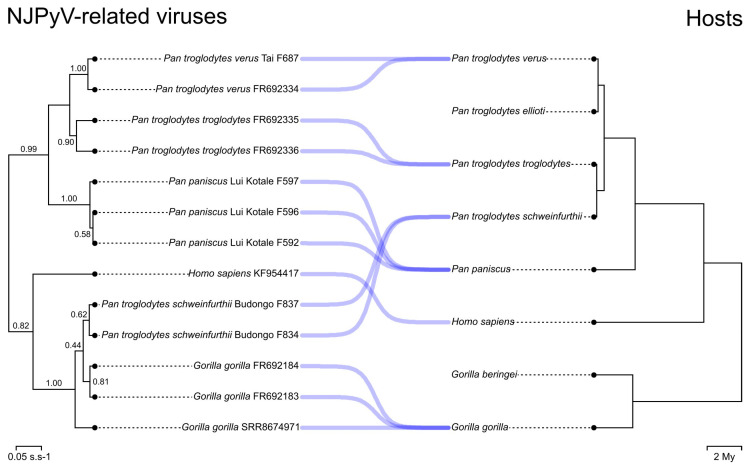
Tanglegram of NJPyV-related viruses and hominine phylogenetic trees. The NJPyV-related virus phylogenetic tree is the MCC tree derived from Bayesian analyses run under an uncalibrated molecular clock model. Branch support is given by posterior probability (numbers above branches). Scales of the NJPyV-related virus and host trees are in substitution per site and million years, respectively.

**Figure 2 viruses-15-02248-f002:**
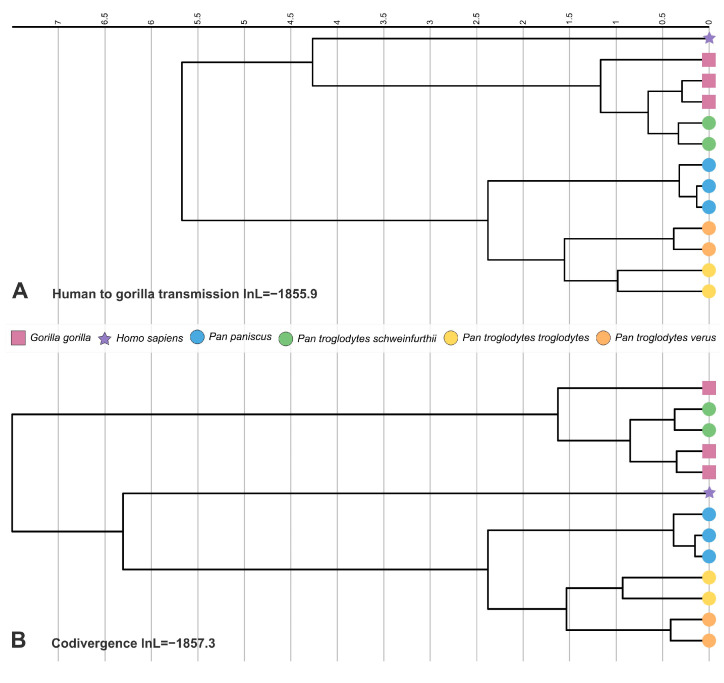
Time trees of NJPyV-related viruses. These NJPyV-related virus phylogenetic trees are the MCC trees derived from Bayesian analyses run under calibrated molecular clock model assuming human to gorilla transmission (**A**) and codivergence (**B**). Scale is in million years.

**Table 1 viruses-15-02248-t001:** Samples used for this study and PCR screening results.

Species/Subspecies	Location (Country)	Number of Samples	Number of Positives (Number of Individuals ^#^)
*Pan troglodytes verus*	Tai National Park (Côte d’Ivoire)	153	4 (3)
*Pan troglodytes troglodytes*	Loango National Park (Gabon)	24	0
*Pan troglodytes schweinfurthii*	Kibale National Park (Uganda)	70	0
*Pan troglodytes schweinfurthii*	Budongo Forest (Uganda)	3	2
*Pan paniscus*	Salonga National Park (DRC *)	10	3 (≥2)
*Pan paniscus*	Malebo (DRC *)	25	4
*Gorilla beringei beringei*	Bwindi Impenetrable National Park (Uganda)	9	0
*Gorilla beringei graueri*	Kahuzi-Biega National Park (DRC *)	25	0
*Gorilla gorilla gorilla*	Loango National Park (Gabon)	10	0

* Democratic Republic of the Congo; ^#^ if known.

**Table 2 viruses-15-02248-t002:** Node age estimates used for calibrated models.

tMRCA ^#^	Perelman et al., 2010 [23] (Mean [95%] HPD ^$^)	Normal Prior * (mean+/−sd [Central 95%])	Kuderna et al., 2023 [24] (Mean [95%] HPD ^$^)	Normal Prior * (Mean+/−sd [Central 95%])
*Homo* + *Pan* + *Gorilla*	8.30 [6.58–10.07]	8.30+/−0.90 [6.54–10.06]	10.12 [8.79–11.24]	10.12+/−0.69 [8.77–11.47]
*Homo* + *Pan*	6.60 [5.40–7.96]	6.60+/−0.70 [5.23–7.97]	8.01 [6.91–8.96]	8.01+/−0.56 [6.91–9.11]
*Pan*	2.17 [1.28–3.21]	2.17+/−0.52 [1.15–3.19]	2.39 [1.98–2.81]	2.39+/−0.22 [1.96–2.82]
*Pan troglodytes*	1.24 [0.66–1.90]	1.24+/–0.32 [0.61–1.87]	NA	NA

# time to the most recent common ancestor; * prior distributions used to model node ages; $ highest posterior density interval.

**Table 3 viruses-15-02248-t003:** Marginal likelihood and transmission event date estimates of molecular clock models.

		No Calibration MLE *	Calibration MLE * (2lnBF_cal/no cal_ ^#^)		Transmission Event Date Mean in Million Years [95% HPD ^&^]	
			Perelman et al., 2010 [23]	Kuderna et al., 2023 [24]	Perelman et al., 2010 [23]	Kuderna et al., 2023 [24]
Model	Codivergence	−1854.2	−1857.3 (−6.2)	−1857.8 (−7.2)	NA	NA
	Incomplete lineage sorting	−1852.3	−1858.0 (−11.4)	−1858.6 (−12.6)	NA	NA
	Gorilla to human		−1857.3 (−10.0)	−1858.0 (−11.4)	4.91 [2.37–7.57]	6.44 [2.75–9.67]
	Human to gorilla		−1855.9 (−7.2)	−1856.8 (−9.0)	4.29 [2.40–6.07]	5.55 [3.08–7.79]

* Marginal likelihood estimate; ^#^ 2lnBF values comparing calibrated and uncalibrated models are considered as positively or negatively decisive when >10 or <−10; ^&^ highest posterior density.

## Data Availability

In addition to being available as Appendix A, new sequences were deposited in GenBank under accessions OR762728-OR762740.

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
