# Peer review of "Evolutionary Insight into the Association between New Jersey Polyomavirus and Humans"

_viruses, 2023, doi:10.3390/v15112248_

Round 1

Reviewer 1 Report

Comments and Suggestions for Authors

Aghebatrafat and colleagues perform a PCR-based survey of close relatives of a mysterious polyomavirus species identified in myopathic samples from a single immunosuppressed human living in New Jersey. The authors also identified a new member of the clade in an SRA dataset representing a gorilla plasma sample. Using the new sequences, the authors report a detailed phylogenetic analysis indicating that the New Jersey polyomavirus has most likely been co-evolving with humans for millions of years. A possible cross-species transmission even involving the ancestors of gorillas and the chimpanzee subspecies schweinfurthii is also estimated to have occurred millions of years ago.

The manuscript presents its complex analyses in a remarkably clear and accessible style. I don't detect any errors, overstatements, or poorly supported claims. The results are interesting and move the field forward in useful ways.

Author Response

Dear Reviewer #1,

We are very grateful for your very positive feedback.

Best regards,

Sebastien Calvignac-Spencer and co-authors

Reviewer 2 Report

Comments and Suggestions for Authors

The manuscript by Aghebatrafat, Calvignac-Spencer et al, is an original research paper aimed to study the phylogenetic of a novel branch of polyomaviruses in gorillas and chimpanzees, in an effort to link them evolutionarily to a recently discovered human polyomavirus, which seems to be limited to one individual.

This work is elegantly designed, clear and follows good scientific practices, with adequate statistical analyses. The data presented is very clear and compelling. The authors demonstrate an early divergence event, possibly millinos of year ago, which resulted in the branch of which this novel polyomavirus in an individual from New Jersey belongs.

The manuscript is very well written and presented; furthermore, the discussion os very thoughtful. The only one minor addition I would suggest is a paragraph explaining the hypothetical origin and hypothetical mode of transmission of the varian detected in the one individual from New Jersey...

This, again will be a highly cited manuscript, which is of general interest, but especially will significantly contribute to the field of polyomaviruses.

Author Response

Dear Reviewer #2,

We are very grateful for your very positive feedback.

We have altered the discussion following your suggestion, adding or extending several sentences/clauses that bring in more information about the single NJPyV case and the possible implications for its origins/transmission (highlighted in blue):

"Since the best-supported placement of NJPyV did not immediately suggest a simple pattern of codivergence (and therefore did not immediately suggest that humans are NJPyV natural hosts), we explored alternative evolutionary scenarios using (molecular clock) model comparisons in a Bayesian framework. Importantly, the two molecular clock models not firmly rejected in these analyses both assumed that the NJPyV lineage infected the human lineage ever since its divergence from African great ape lineages. We conclude that the patterns of genetic diversity of NJPyV/NJPyV-related viruses are, at this stage, compatible with NJPyV being a human-infecting agent. Importantly, this aligns well with the NJPyV patient history, since this person did not report recent international travels, nor contact with domestic animals or nonhuman primates [1]. We acknowledge that further sampling of hominine- and other mammal-infecting PyV may alter the patterns we observed. However, we do not see this eventuality as very likely, considering the already consequent search effort in African great apes.

If one accepts the notion that NJPyV is a human-infecting virus, the apparent absence of NJPyV from human populations remains a conundrum. It seems unlikely that NJPyV would currently be on the brink of extinction in humans, given the ability of its close relatives in African great apes to survive in much smaller fragmented populations. Plausible explanations might include: (i) that NJPyV replication levels in healthy individuals are extremely low, consistent with undetectable viral load in a pretransplant serum sample of the NJPyV patient [1]; (ii) that its main site of replication has been poorly sampled and tested (NJPyV exhibited a broad vascular endothelial cell tropism in the only known case but whether this reflects its tropism in healthy individuals is unknown [1]); and/or (iii) that it has an uneven global distribution and is more prevalent in (or even nearly restricted to) poorly sampled regions of the world, like several recently discovered lineages of pathogens [41,42]. We hypothesize that NJPyV and possibly other human-adapted viruses rarely associated with disease will be detected in the future, as such biases are progressively corrected for."

In addition, we have also followed the suggestion of the handling editor,  replacing all NJPyVr acronyms with appropriately spelled out expressions, and we have obtained Genbank accession numbers for the sequences generated for this study (the sequences will be released as soon as processing by GenBank staff is completed).

Best regards,

Sebastien Calvignac-Spencer and co-authors